# Powder Nano-Beam Diffraction in Scanning Electron Microscope: Fast and Simple Method for Analysis of Nanoparticle Crystal Structure

**DOI:** 10.3390/nano11040962

**Published:** 2021-04-09

**Authors:** Miroslav Slouf, Radim Skoupy, Ewa Pavlova, Vladislav Krzyzanek

**Affiliations:** 1Institute of Macromolecular Chemistry of the Czech Academy of Sciences, Heyrovsky Sq. 2, 162 06 Prague 6, Czech Republic; pavlova@imc.cas.cz; 2Institute of Scientific Instruments of the Czech Academy of Sciences, Kralovopolska 147, 612 64 Brno, Czech Republic; ras@isibrno.cz

**Keywords:** nanoparticle analysis, powder nanobeam electron diffraction, 4D-STEM/PNBD

## Abstract

We introduce a novel scanning electron microscopy (SEM) method which yields powder electron diffraction patterns. The only requirement is that the SEM microscope must be equipped with a pixelated detector of transmitted electrons. The pixelated detectors for SEM have been commercialized recently. They can be used routinely to collect a high number of electron diffraction patterns from individual nanocrystals and/or locations (this is called four-dimensional scanning transmission electron microscopy (4D-STEM), as we obtain two-dimensional (2D) information for each pixel of the 2D scanning array). Nevertheless, the individual 4D-STEM diffractograms are difficult to analyze due to the random orientation of nanocrystalline material. In our method, all individual diffractograms (showing randomly oriented diffraction spots from a few nanocrystals) are combined into one composite diffraction pattern (showing diffraction rings typical of polycrystalline/powder materials). The final powder diffraction pattern can be analyzed by means of standard programs for TEM/SAED (Selected-Area Electron Diffraction). We called our new method 4D-STEM/PNBD (Powder NanoBeam Diffraction) and applied it to three different systems: Au nano-islands (well diffracting nanocrystals with size ~20 nm), small TbF_3_ nanocrystals (size < 5 nm), and large NaYF_4_ nanocrystals (size > 100 nm). In all three cases, the STEM/PNBD results were comparable to those obtained from TEM/SAED. Therefore, the 4D-STEM/PNBD method enables fast and simple analysis of nanocrystalline materials, which opens quite new possibilities in the field of SEM.

## 1. Introduction

Scanning electron microscopy (SEM) and transmission electron microscopy (TEM) are well-established imaging and analytical techniques for characterization of surfaces and thin samples, with numerous applications in many fields of material and life sciences. Typical SEM microscopes use electrons with an energy of 30 keV and below, while typical TEM microscopes operate in a higher energy range between 60 and 300 keV. In both SEM and TEM, it is possible to detect transmitted electrons by means of a technique called scanning transmission electron microscopy (STEM) [1]. The SEM and STEM images are recorded sequentially, i.e., a thin electron beam scans a selected rectangular area on a specimen and the electronic signal from a detector is attached to the beam position. In this way, we detect secondary electrons (SE), backscattered electrons (BSE), as well as transmitted electrons (TE) by means of bright field (BF), annular dark field (ADF), or high-angle annular dark field (HAADF) detectors. The resulting STEM/BF, STEM/ADF, and STEM/HAADF micrographs are two-dimensional (2D-STEM) in the sense that one XY-position on the sample gives one signal on the detector [1,2].

Further development in the field of STEM imaging headed towards multi-segmental annular detectors and, subsequently, to 2D pixel array detectors (also known as pixelated detectors). Multi-segmental STEM detectors consist of a central segment in the BF region and several annular (or semi-annular) segments in ADF and HAADF regions (the typical number of segments is between 4 and 10, depending on the manufacturer [3,4,5]). In this case, several signal values are related to one beam position and, as a result, several images can be taken during each frame scan (this is occasionally called multidimensional microscopy, where the additional dimension is the angular resolution [6]). For finer angular resolution, the STEM detector could be further modified by introducing the scattered beam limiting apertures [7]. The traditional 2D pixel array detectors employed in STEM imaging are cameras based on CCD (charge-coupled device) or CMOS (complementary metal oxide semiconductor) technology, but these devices suffer from somewhat lower speed and/or higher noise. The modern 2D array detectors are DED (direct electron detectors), and these devices exhibit higher speed, lower noise, and smaller size than the traditional CCD and CMOS cameras coupled with various scintillators [8,9].

The pixelated STEM detectors record intensity of the scattered electrons at given primary beam position as a function of their direction or, in other words, they record a 2D diffraction pattern at current beam position. If the scanning beam is narrow (and not convergent), the probe size is small and the recorded patterns correspond to nanobeam diffraction (NBD) known from TEM (we note that the classical TEM/NBD method is performed with at fixed beam position [10]). As the diffraction pattern is recorded at each primary beam position on the specimen, we get a four-dimensional (4D) data cube (a 2D array of 2D diffraction patterns) and the technique is referred to as 4D-STEM. The 4D-STEM technique was introduced in the field of TEM and dedicated high-energy STEM microscopy [11], and, despite being quite new, it has already found several applications, such the mapping of electrostatic fields in 2D semiconductors [12,13], electron ptychographic diffractive imaging [11,14], mapping of structure of amorphous thin films by means of the pair distribution function analysis [15], or mapping of orientation of organic nanocrystal grains [16]. In the field of SEM microscopy, the pixelated detectors based on fast DED cameras has been commercialized very recently [17]. As a result, the 4D-STEM techniques in SEM microscopes are less common, just a few examples were found in the available literature, and almost all methods have been based on traditional CCD or CMOS detectors [7,18,19,20,21].

From the point of view of everyday routine use, all above-listed applications of 4D-STEM microscopy have two limitations: At first, the great majority of 4D-STEM methods were developed and tested on TEM. Nevertheless, even simplified versions of these methods would substantially enhance the analytical possibilities of SEM microscopes, where the classical diffraction methods, such as selected-area electron diffraction (TEM/SAED), are unavailable. Second, all above-listed 4D-STEM methods focus on analysis of individual diffraction patterns, which is usually complex and requires detailed crystallographic knowledge together with specialized software. The complexity consists in that the analyzed three-dimensional (3D) nanocrystalline materials are (in great majority of real applications) oriented randomly and, as a result, we record randomly oriented sections of their 3D diffractograms.

In this work, we introduce a novel, fast, and simple 4D-STEM diffraction technique, which can be employed in any SEM microscope equipped with a pixelated STEM detector. The method does not try to analyze individual diffraction patterns from every single nanocrystal and/or location but combines the whole 4D-STEM dataset into one composite diffraction pattern that is completely analogous to TEM/SAED diffractogram of a polycrystalline sample. We called the new method 4D-STEM/PNBD (Powder NanoBeam Diffraction), as the resulting diffractogram is a combination of the individual NBD patterns. The great advantage of our simplified approach consists in that the 4D-STEM/PNBD polycrystalline diffraction patterns are much easier to process than the whole dataset of individual diffractograms. This makes the method accessible to all experienced users, not only to professional crystallographers. Moreover, the processing of 4D-STEM/PNBD polycrystalline diffraction patterns does not require specialized, heavy-weight crystallographic packages. The summation of the individual diffractograms can be performed in a few minutes by means of quite simple and easy-to-adjust scripts in freeware Python programming language (these scripts are currently available upon request from the first author). The processing of the final 4D-STEM/PNBD diffractogram can be performed by any small, simple, and stand-alone program for analysis of TEM/SAED powder diffraction patterns (such as ProcessDiffraction [22]). The new 4D-STEM/PNBD method has its own specific limitations, which are exemplified and discussed below, but it definitely opens quite new possibilities for SEM users. The recently developed pixelated STEM detectors can be easily installed in any modern SEM microscope. The SEM microscope with pixelated detector and quite straightforward 4D-STEM/PNBD technique can be used with ease, not only for imaging and spectroscopy, but also for diffraction analysis of polycrystalline samples.

## 2. Materials and Methods

### 2.1. Samples

Three nanocrystalline samples with various average crystal size were selected for testing our new diffraction technique. In all three cases, the nanocrystals were deposited on a standard TEM copper grid coated with an electron-transparent carbon film. The first sample was well diffracting Au nano-islands layer (size ~20 nm), the second sample contained small TbF_3_ nanocrystals (size < 5 nm), and the third sample consisted of NaYF_4_ nanocrystals (size > 100 nm). The preparation of the samples was described in detail in our previous studies [23,24,25]. Briefly, the medium-sized Au nano-islands were used as a nucleating agent for polypropylene [23], the small TbF_3_ nanocrystals (with admixture of Gd^3+^, Yb^3+^, and Nd^3+^) were designed as multimodal contrast agents for down- and up-conversion luminescence, magnetic resonance imaging, and computed tomography [24], and the large NaYF_4_ nanocrystals (with admixture of Yb^3+^ and Er^3+^) were synthesized as water-dispersible, stable, light up-conversion nanoparticles for biomedical applications [25].

### 2.2. TEM Characterization

Morphology, elemental composition, and crystalline structure of all three samples (Au, TbF_3_, and NaYF_4_) were thoroughly characterized by means of TEM (microscope Tecnai G2 Spirit Twin; FEI, Czech Republic; accelerating voltage 120 kV). The morphology of the nanocrystals was visualized by standard bright-field imaging (TEM/BF). Elemental composition was assessed from energy-dispersive X-ray analysis (TEM/EDX). The expected crystalline structure (face-centered cubic for Au, orthorhombic structure of TbF_3_, and hexagonal structure of NaYF_4_) was confirmed using selected-area electron diffraction (TEM/SAED). At given experimental conditions, all samples yielded polycrystalline, powder electron diffraction patterns. The experimental 2D-diffraction patterns were processed and converted to radially averaged 1D-diffraction profiles by means of the ProcessDiffraction program (freeware program, version version 8.7.1, ref. [22]). Additional experimental and data processing details can be found in the studies dealing with Au, TbF_3_, and NaYF_4_ preparation [23,24,25].

### 2.3. Calculation of PXRD Diffraction Patterns

The experimental TEM/SAED diffraction patterns were compared with theoretical, calculated powder X-ray diffraction patterns (PXRD). The crystal structures were downloaded from the Crystallography Open Database [26]. The calculation of theoretical PXRD patterns was performed with the program PowderCell (freeware program, version 2.4, ref. [27]).

### 2.4. STEM Measurements Including the 4D-STEM/PNBD Method

#### 2.4.1. SEM Microscope with Pixelated Detector

This article introduces the 4D-STEM/PNBD method, that yields powder electron diffraction patterns by means of an arbitrary SEM microscope equipped with a pixelated STEM detector. We used a focused ion beam scanning electron microscope Helios G4 (FIB-SEM microscope; Thermo Fisher Scientific, Waltham, MA, USA) equipped with pixelated STEM detector T-pix (Thermo Fisher Scientific, Waltham, MA, USA), which was based on Timepix technology [28]. Additional equipment of the microscope comprised an annular STEM3+ detector, which was used for high-resolution bright field images.

The pixelated STEM detector was based on a Si sensor, composed of 256 × 256 pixels (pixel size 55 µm; detector active area of 14 × 14 mm; declared detection efficiency at 30 keV higher than 90%) and mounted at a distance of 40 mm from the pole piece. Therefore, it covered scattering angles from 180 to 200 mrad, depending on the working distance of a sample (computed for 2 and 6 mm). The in-chamber situation during the experiment is shown in Figure 1. When 4D-STEM data were taken, the 2D-STEM detector was inserted below the sample holder and the annular STEM3+ detector was retracted.

#### 2.4.2. Principle of 4D-STEM/PNBD Method

The principle of 4D-STEM/PNBD method is shown in Figure 2. The figure is based on real images from the first testing sample (Au nano-islands, Section 2.1).

The 4D-STEM/PNBD measurements comprise three basic steps: In the first step, we acquire a standard STEM bright-field image (STEM/BF, Figure 2a) and define the rectangular area from which we will collect the individual diffractograms (rectangular array of red points on Figure 2a). In the second step, we collect diffractograms from the selected area (Figure 2b, each red point in Figure 2a corresponds to one diffraction pattern). We note that the individual diffractograms contain spots as they come from small locations containing usually one or just a few crystallites. In the third step, the individual spotty diffraction patterns are summed. The summation of randomly oriented diffraction patterns from the previous step results in a powder diffraction pattern (Figure 2c) that displays typical diffraction rings.

For the collection of the initial STEM/BF micrograph (step 1 of the 4D-STEM/PNBD method, Figure 2a), we can use both pixelated detectors or, more conveniently, the standard STEM detector. Then, the control software of the pixelated detector is employed in defining the rectangular area on the STEM/BF image (the red points in Figure 2a), from which the individual diffraction patterns will be collected.

For the collection of the individual diffraction patterns (step 2 of the 4D-STEM/PNBD method, Figure 2b), we used the pixelated detector and the spot mode of the internal scanning unit. The distance between neighboring spots is set according to typical crystal size of the investigated sample in order to minimize multiple data collections from one crystal (a suppression of data redundancy as multiple measurements of the same crystal would yield almost identical diffraction patterns). The size of the scanned area is then readily calculated from the intended number of points and the step size. In the case of big crystals and big scanning steps, there is a limitation—a shift of the center of the diffraction pattern caused by too-large beam shift during scanning. This misalignment of the individual diffractograms would lead to a blurred powder diffractogram in the final step. In fact, the beam shift could be corrected during summation of diffractograms, but we avoided this problem by using maximal scan area below 10 µm. At this setting, the beam shift was lower than the size of one pixel on the detector. The 4D datasets (2D-array of diffractograms that are also 2D-arrays) were captured, with the settings shown in Table 1. The beam energy (30 keV), beam current (25 pA), and dwell time (100 µs) were the same in all experiments. The microscope optics was set to field free mode.

For the summation of the individual diffraction patterns (step 3 of the 4D-STEM/PNBD method, Figure 2c), we prepared a program package called STEMDIFF. It is a set of scripts in Python programing language which can be easily modified and adjusted for a given dataset. The detailed description of the scripts is given below in the Results Section, as the STEMDIFF package is one of the significant outputs of this work and, more importantly, the data processing is closely associated with the final results. In brief summary, the scripts visualize the individual diffractograms (which are stored as binary files containing intensities measured at each pixel on the 2D-STEM detector), then the scripts can filter out diffractograms without diffraction spots (that only increase background noise, but carry no useful information); subsequently, the scripts can perform the final summation (for both unfiltered and filtered datasets), and finally, the scripts can convert the resulting 2D-diffraction image to a one-dimensional (1D)-diffraction profile (radial averaging).

The final diffractogram (Figure 2c) can be processed with arbitrary software for powder diffraction pattern analysis. In our case, we analyzed the 4D-STEM/PNBD patterns with the same software that was used for the analysis of TEM/SAED patterns: ProcessDiffraction program (for the processing of experimental powder electron diffraction patterns [22]; freeware, version 8.7.1) and PowderCell program (for the calculation of theoretical powder X-ray diffraction patterns [27]; freeware, version 2.4).

## 3. Results

### 3.1. Results of 4D-STEM/PNBD Method

#### 3.1.1. Au Nano-Islands

The results of TEM and STEM analysis of Au nano-islands are summarized in Figure 3. The TEM/BF and STEM/BF micrographs (Figure 3a,c) confirmed that the average size of the irregular-shaped nano-islands was around 20 nm. The TEM/SAED and 4D-STEM/PNBD diffraction patterns (Figure 3b,d) were quite similar, although the diffraction rings on TEM/SAED were sharper. For the sake of easier visual comparison, both diffractograms were resized and post-processed with ImageJ [29] so that the overall magnification and the diffraction intensities were similar. For the final calculations (Figure 3e), the original, unmodified diffractograms were used in order to get correct results.

We note that the TEM/SAED image shows a beam-stopper (black strip in Figure 3b), which had to be used during TEM measurements, as the high-intensity primary beam could damage the sensitive TEM digital camera. For the STEM pixelated detector, the beam-stopper was not necessary, because the detector could withstand the very high doses and overflown intensity in the central spot, corresponding to the primary beam. The 4D-STEM/PNBD diffractogram in Figure 3d was obtained from the smaller of the two datasets measured for Au nano-islands (Table 1, dataset: Au small). Despite this fact, the quality of the diffraction data was sufficient. The influence of dataset size on the 4D-STEM/PNBD results is discussed in more detail in Section 3.3.1.

The 4D-STEM/PNBD and TEM/SAED diffraction patterns (Figure 3b,d) were converted to 1D-diffractograms (by means of the ProcessDiffraction program [22]) and compared with the theoretical PXRD pattern of Au (calculated with the PowderCell program [27]). This final quantitative comparison is shown in Figure 3e. Very good agreement between both experimental methods (PNBD and SAED) and theoretical calculation (PXRD) was achieved. Both positions and intensities of the diffraction peaks were similar. The small discrepancies could be attributed to the intrinsic differences between X-ray and electron diffraction, as well as to experimental errors. The width of the diffraction peaks increased in the following order: PXRD < TEM/SAED < 4D-STEM/PNBD. This effect was associated mostly with the small size of the Au nano-islands. Theoretical PXRD calculation was based on ideal infinite Au crystal, while TEM/SAED and 4D-STEM/PNBD measurements were performed on real Au nanocrystals, whose size was around 20 nm. According to the Scherrer equation (τ = *K*λ/(β⋅cosθ), where τ is the average size of the crystalline domain, *K* is the dimensionless shape factor ≈ 0.9, β is the diffraction peak broadening at half the maximum intensity (FWHM), and θ is the diffraction angle), the width of a diffraction peak (β) is indirectly proportional to the size of crystallites (τ). The approximate Scherrer equation holds quite well for both X-ray and electron diffraction [30]. The reasons why the 4D-STEM/PNBD diffraction peaks were even broader than TEM/SAED diffraction peaks and how this could be improved in the future are discussed below (Section 4). In any case, even the 4D-STEM/PNBD diffraction pattern with the broadest diffraction peaks corresponded to the Au crystal structure.

#### 3.1.2. TbF_3_ Nanoparticles

The results of TEM and STEM analysis of TbF_3_ nanocrystals are compiled in Figure 4. The TEM/BF and STEM/BF micrographs (Figure 4a,c) confirmed the small size of the TbF_3_ nanocrystals (they appear as small black dots with size < 5 nm in leaf-shaped agglomerates). The overall shape of the TEM/SAED and 4D-STEM/PNBD diffraction patterns (Figure 4b,d) was similar, and the radially averaged intensities were in good agreement with the theoretically calculated PXRD pattern (Figure 4e). The average width of diffraction peaks increased in the order PXRD < TEM/SAED < 4D-STEM/PNBD, like in the previous case of Au nano-islands. The reasons for the observed broadening were the same in both cases, but the effect seemed to be even stronger in the case of smaller TbF_3_ nanoparticles with lower diffraction power.

The intensities of TEM/SAED and 4D-STEM diffraction peaks were similar (red and black lines in Figure 4), but they somewhat differed from the theoretically calculated PXRD intensities (blue line in Figure 4). This seeming discrepancy could be attributed to the preferred orientation of the small TbF_3_ nanocrystals, as already explained in our previous study that was focused on the synthesis, TEM characterization, and chemical modifications of GdF_3_ and TbF_3_ nanoparticles [24]. Briefly, the theoretical PXRD diffraction pattern was calculated for random orientation of crystals, while the experimental TEM/SAED (as well as 4D-STEM/PNBD) diffractograms were measured from a thin layer of nanocrystals deposited on the electron transparent carbon film. The TbF_3_ formed thin nanoplatelets, most of which laid on their small facets oriented in such a way that their shortest unit cell parameter, *c*, was parallel with the electron beam [24]. This corresponded to the preferred orientation of the nanocrystals with zone axis [*uvw*] = [001]. According to Weiss zone law (WZL: *hu* + *kv* + *lw* = 0, where *h*, *k*, *l* are diffraction indexes and *u*, *v*, *w* = [001] are the indexes of the zone axis [31]), the strongest diffraction peaks should be of the type [*hk*0], i.e., their last diffraction index should be zero (because in our case, the WZL takes a simple form: *hu* + *kv* + *lw* = *h0* + *k0* + *l1* = *l* = 0). Indeed, the *hk*0 diffraction peaks from TEM/SAED and 4D-STEM/PNBD were stronger than corresponding calculated PXRD diffraction peaks, while the intensive *hkl* PXRD diffractions (such as 111, 121, and 232) exhibited lower or even negligible intensity in the experimental SAED and PNBD patterns.

#### 3.1.3. NaYF_4_ Nanoparticles

The results of TEM and STEM analysis of NaYF_4_ nanoparticles are shown in Figure 5. The large NaYF_4_ microcrystals looked transparent for high-energy electrons in TEM/BF imaging at 120 keV (Figure 5a, gray hexagons showing internal structure in the form of black stripes that could be interpreted as bend contours [32]), but substantially less transparent for low-energy electrons in STEM/BF imaging at 30 keV (Figure 5b, black hexagons with no internal morphology). Consequently, the TEM/SAED diffractogram (Figure 5b) showed sharp diffraction rings, while the 4D-STEM/PNBD diffractogram (Figure 5d) displayed broad rings of low contrast with respect to the background. Nevertheless, even the broad 4D-STEM/PNBD diffraction peaks showed good agreement with the TEM/SAED after radial averaging (Figure 5e). The positions of 4D-STEM/PNBD and TEM/SAED diffraction peaks in Figure 5e corresponded very well to the calculated PXRD pattern, but the intensities were somewhat different. The reason was a preferred orientation: NaYF_4_ formed flat hexagons laying mostly on their 001 facets (like the nanoplatelets of TbF_3_). Consequently, the nanocrystals exhibited a strong preferred orientation with zone axis [*uvw*] = [001] and the strongest experimentally observed diffraction peaks were of *hk0* type (bold font in Figure 5e), while the general *hkl* diffractions with l ≠ 0 exhibited low or even negligible intensity (more details can be found in our previous study [25]).

### 3.2. STEMDIFF: Program Package for Convenient Processing of 4D-STEM/PNBD Data

The program package STEMDIFF was created within this work for fast, easy, and convenient processing of 4D-STEM/PNBD data. It is worth remembering that each position of the beam in the scanned rectangular 2D area produces a 2D-array of intensities on the pixelated detector. The TbF_3_ dataset, which is used as an example in this section, contained 100 × 80 = 8000 positions (Table 1), and each position corresponded to a 256 × 256 array of detected intensities, resulting in >500 million numbers in 8000 files with total size >1 GB. This simple calculation illustrates that the manual processing of 4D-STEM/PNBD datasets was impossible and suitable software had to be created.

The STEMDIFF package is a set of scripts in Python programing language. The STEMDIFF package relies on fast, well-established Python modules (NumPy for fast data treatment by means of array computing, matplotlib for plotting, and Pillow and scikit.image for image operations). All modules are part of many Python distributions (such as WinPython, which was used in this work). Therefore, usage of STEMDIFF is extremely easy: after installing a suitable freeware Python distribution that contains the necessary modules (such as WinPython or Anaconda), it is enough to download the scripts (at the moment they are available upon request from the first author of this work) and apply them to a given dataset. Another advantage of our approach relying on standard scientific modules consists in that the underlying algorithms of the modules are highly optimized, typically written in C or Fortran and, as a result, the STEMDIFF scripts are very fast. Even the processing of the largest datasets (>2.5 GB of data) took less than 3 min with a common modern computer.

The scheme of 4D-STEM/PNBD data processing with the STEMDIFF package is shown in Figure 6. The input is the set of data files (Figure 6a). Each file contains a 256 × 256 array of intensities in the form of a binary file. In the simplest case, it is possible to sum the intensities in all files (script *sum-all.py*) and save the resulting file after suitable rescaling as a grayscale diffraction pattern image (script *rescale.py*, Figure 6b). Then, it is possible to calculate the radially averaged intensity (script *rad-ave.py*, Figure 6c) in order to estimate the quality of the processed data, i.e., the intensity of diffraction peaks with respect to the background. In the case of the TbF_3_ dataset in Figure 6, the direct summation of all data files yielded very weak diffractions that were not suitable for further calculations. This was caused by the fact that the TbF_3_ particles were small (low-intensity diffractions) and agglomerated (numerous beam positions in the rectangular scanned area yielded diffractograms of the supporting carbon film that contained just central spot and background noise). For such cases, the STEMDIFF package enables the automatic filtering of the dataset and summation of the files containing strong diffraction spots, while the files containing noisy information (i.e., only the central spot or a just a few weak diffraction spots) are excluded. At first, the dataset is visualized by means of script *inspect-data.py* (Figure 6d), which traverses through the data files, shows each file as a grayscale diffraction image, and displays its calculated Shannon entropy value, *S* [33]. The value of *S* tends to be higher for diffractograms containing sharp and intensive diffraction spots and lower for diffractograms containing only the central spot and perhaps a few weak diffraction spots. The user goes through the dataset and determines a suitable threshold value of *S*. This value is inserted as an input to the following script (*sum-selected.py*), which then sums only the data files with strong diffraction spots (i.e., with *S* > user-defined threshold). After rescaling (script *rescale.py*), the final diffractogram is saved as a grayscale image (Figure 6e) and the quality of the result can be estimated by visualizing its radially averaged intensity (script *rad-ave.py*, Figure 6f). It is evident that this entropy-based filtering can improve the result significantly (compare Figure 6c,f). In fact, the 4D-STEM/PNBD diffractogram after filtering with *S* > 8.6 (Figure 6e) was used for the final calculations that showed the very good agreement with the TEM/SAED experiment and PXRD calculation (Figure 4e). To summarize, the STEMDIFF package reads the raw data files from a pixelated detector (Figure 6a) and yields two outputs: (i) the powder diffractograms in the form of grayscale images (Figure 6b,e), which can be further processed by arbitrary software, in complete analogy with TEM/SAED diffraction patterns, and (ii) the supplementary output are the radially averaged intensities (Figure 6c,f), which can be used for the fast estimate of the dataset quality. The user interface of the STEMDIFF package is shown in Figure 7.

### 3.3. Influence of Selected Parameters on the Quality of 4D-STEM/PNBD Results

#### 3.3.1. Dataset size

The effect of the dataset size (i.e., the number of files in the processed dataset) on the quality of final 4D-STEM/PNBD diffraction patterns was small. The radially averaged intensity of 4D-STEM/PNBD diffractograms showed negligible changes with increasing size of the dataset (Figure 8). All data in Figure 8 come from dataset *Au big* (Table 1), that contained as many as 40,000 individual diffraction data files. Both diffractograms (Figure 8a,b) and radially averaged intensities (Figure 8c) were calculated by means of STEMDIFF scripts (Figure 6). The diffractograms were produced by *sum-all.py* and *rescale.py* scripts (Figure 6b), while the radially averaged diffraction intensity was calculated by the *rad-ave.py* script (Figure 6c). The 4D-STEM/PNBD diffractogram was obtained by summation of the whole dataset (40,000 files, Figure 8a) and showed well-averaged, smooth, and compact diffraction rings, while the diffractogram obtained by summation of just the first 100 files (Figure 8b) displayed less compact diffraction rings composed of partially overlapping spots. Nevertheless, the overall intensities of the rings looked very similar and the radially averaged intensities were almost indistinguishable, regardless of the dataset size, on the condition that the number of files ≥100 (Figure 8c). This explains why we could obtain high-quality 4D-STEM/PNBD data even with the smaller dataset, *Au small* (Table 1, Figure 3), and why the increase in dataset size yielded almost identical results (not shown here for the sake of brevity).

#### 3.3.2. Dataset Filtering

The effect of the dataset filtering (i.e., the removal of files that contain none or weak diffraction spots from the processed dataset) was strong and important. The results for all three samples are summarized in Figure 9. The figure shows radially averaged intensities, which were calculated by the STEMDIFF package (scripts *sum-all.py*, *sum-selected.py*, *rescale.py,* and *rad-ave.py*), as shown schematically in Figure 6 and described in Section 3.2.

The impact of filtering is closely connected with the average nanocrystal size, the diffraction power of the nanocrystals with respect to supporting carbon film, and the overall sample morphology. The first sample, Au nano-islands, represented an optimal specimen for the 4D-STEM/PNBD method: the nano-islands diffracted electrons strongly due to high atomic number of gold (this resulted in strong diffraction spots), their size was close to ideal (neither too small and weak-diffracting, nor too big and high-absorbing), and they cover the carbon film in a homogeneous layer (which minimized the amount of data files containing only background noise). Consequently, the Au dataset exhibited intensive diffractions even without any modification (Figure 9a, dotted line) and the filtering just somewhat improved the data quality (Figure 9a, full lines). The full dataset contained 40,000 files, the filtered dataset without the files with weak diffraction spots (i.e., containing just files with *S* > 7.7) had 22,241 files, and the filtered dataset containing just the files with strong diffraction spots (*S* > 8.0) comprised 3485 files. The results for dataset *Au small* (not shown in Figure 9) were almost identical to *Au big* (as indicated by Figure 8): the original *Au small* dataset contained 2000 files, and the filtered datasets with S > 7.7 and S > 8.0 were composed of 1271 and 545 files, respectively. The *Au small* dataset with S > 8.0 was employed in the final calculation and the results were in very good agreement with both the TEM/SAED experiment and PXRD calculation, as evidenced in Figure 3.

The small TbF_3_ nanocrystals represented a more difficult sample for the 4D-STEM/PNBD method: their diffraction power was lower in comparison with Au and not much higher in comparison with the supporting carbon film (due to the smaller size, lower average atomic number, and lower density of nanocrystals). Moreover, the TbF_3_ nanocrystals tended to form agglomerates, which led to the empty areas on the supporting carbon film, and the diffractograms of the carbon film just increased the background noise. As a result, the unfiltered dataset (Figure 9b, dotted line) showed just small and broad diffraction peaks, which were hard to distinguish from the background. These peaks were almost unusable for further processing. The entropy-based filtering (*S* > 8.2; Figure 9b, yellow line) that removed the files with no diffractions (empty spaces on carbon film) together with the files containing just very weak diffraction spots (edges of the agglomerates) yielded a much better dataset, where the diffraction peaks were significantly enhanced with respect to the background. The best results were obtained with dataset containing only data files with strong diffraction spots (*S* > 8.6; Figure 9b, red line). The number of files in the original dataset (8000) decreased to 3959 and 1010 after the first and the second filtering, respectively. The best dataset containing files with strong diffraction spots (*S* > 8.6) was used for the final comparison with TEM/SAED and PXRD results and good agreement was achieved as well, as documented in Figure 4.

The large NaYF_4_ nanocrystals were the most difficult sample for 4D-STEM/PNBD: the crystals were transparent for 120 keV electrons in TEM/SAED (Figure 4b), but almost opaque for 30 keV electrons in 4D-STEM. Consequently, the amount of diffractograms with strong diffraction spots was rather limited: the empty spaces in between the crystals produced just scattering from carbon, and too-thick crystals exhibited weak diffraction spots due to absorption, multiple scattering, and backscattering effects. We decided to combine datasets from several locations in order to get a sufficient amount of diffraction patterns for filtering and summation (Table 1, dataset NaYF_4_, 10 locations, 20,000 files). The 4D-STEM/PNBD diffractogram calculated from the dataset without filtering showed virtually no peaks, because a great majority of the diffractograms contained a lot of background noise (Figure 9c, dotted line). Fortunately, the entropy-based filtering worked even in this difficult case (Figure 9c, full lines). The filtered datasets with *S* > 8.3 and *S* > 8.7 contained just 4503 and 524 files, respectively. The background subtraction for the best dataset (*S* > 8.7) was not easy due to low-intensity, broad diffraction peaks (Figure 9c, red line), but the final 4D-STEM/PNBD radial profile exhibited surprisingly good agreement with both the TEM/SAED experiment and PXRD calculation, as illustrated in Figure 5.

It is worth mentioning that the dataset filtering enhanced the diffraction peaks for all three systems (Au nano-islands, small TbF_3_ nanocrystals, and large NaYF_4_ nanocrystals) in the whole range of radial distances and diffraction intensities (Figure 9). In other words, the dataset filtering enhanced all diffraction peaks regardless of their position and intensity, without eliminating the weaker diffraction peaks and biasing the results. This was documented not only in Figure 9 (no vanishing peaks after dataset filtering), but also in Figure 3, Figure 4 and Figure 5 (the very good agreement among the 4D-STEM/PNBD results based on the filtered datasets, independent TEM/SAED experiments, and PXRD calculations).

## 4. Discussion and Conclusions

### 4.1. Originality of 4D-STEM/PNBD Method

The 4D-STEM/PNBD method was developed with the idea that the application of the 4D-STEM technique in a SEM microscope should be as simple as possible. As documented in the Introduction Section, the 4D-STEM was originally introduced in the field of TEM and the number of papers about 4D-STEM in SEM is rather limited. Moreover, a great majority of the SEM papers dealing with the 4D-STEM method are based on CCD or CMOS detectors (slower, less efficient, and more difficult to install inside the microscopic chamber due to the fact that they have to be coupled with a scintillator), while our contribution is based on a DED-based pixelated STEM detector (faster and easy to install in any modern SEM microscope with a suitable port, because the size and geometry of the pixelated STEM detector are comparable to a conventional STEM detector).

The main difference between the 4D-STEM/PNBD method and all other 4D-STEM techniques described in the literature consists in that that 4D-STEM/PNBD intentionally avoids processing of the individual diffractograms (that requires special crystallographic software and knowledge) and converts the 4D-STEM data cube into one simple powder electron diffraction pattern (that can be processed easily with exactly the same simple programs that are used for TEM/SAED powder diffractograms). Nevertheless, the possibility to convert the whole 4D-STEM dataset into one powder electron diffractogram did not escape the attention of other researchers completely, although their studies were focused on different 4D-STEM data processing strategies. The detailed review dealing with basic principles and applications of 4D-STEM methods in TEM [11] summarized various possibilities of how to process 4D-STEM datasets and how to employ the results for virtual imaging, structure classification, orientation mapping, strain mapping, characterization of non-crystalline materials, ptychographic reconstructions, etc. All above-listed methods rely heavily on analysis of individual 2D datasets within the 4D-STEM data cube, but the possibility to average the diffractograms from selected locations is briefly mentioned as one of the additional options. This is quite logical, because in TEM, the powder electron diffraction patterns from selected areas can be obtained readily with the well-established TEM/SAED method, without additional complexities connected with the 4D-STEM technique. A very recent electronic book about applications of 4D-STEM in SEM [7] concentrates on analysis of the individual single crystal diffraction patterns in a system equipped with both a conventional STEM detector and p-STEM (programmable detector based on phosphor screen and CMOS camera for recording diffractograms and photomultiplier tube for STEM imaging). Powder electron diffraction patterns are briefly mentioned, but they are obtained by completely different and relatively complex techniques employing a conventional STEM detector combined with a system of masks and annular apertures. Yet another very recent publication by Schweizer at al. [21] describes a SEM system equipped with a fluorescent screen at the bottom of a specimen chamber and dedicated in vacuo CMOS camera. The system is different, the probability of detecting diffracted electrons is low (2.2% at 30 keV, as estimated by the authors), and the work is focused on the processing of monocrystal diffraction patterns, but the possibility to obtain polycrystalline diffraction patterns like in our case is clearly mentioned.

### 4.2. Current Limitations of the 4D-STEM/PNBD Method

The most important limitation of the current version of the 4D-STEM/PNBD method consists in high background that may be difficult to remove in the case of low-intensity diffraction peaks. We have demonstrated that the influence of background can be suppressed significantly by dataset filtering (Figure 9). Nevertheless, there are some other effects that should be considered and minimized in the future. The primary electron beam with a semi-angle of 0.76 mrad, which was used in this study, covers the detector plane area with the diameter of 60 µm, while the pixel size is just 55 μm. More importantly, the beam is spread by Gaussian beam profile and expanded by the electron scattering of the support carbon layer. The resulting point spread function (PSF) is then convoluted with the diffraction pattern of the measured sample. This leads to the blurring of individual diffraction spots and the increasing of the background noise during the summation of individual patterns. Both theoretical considerations [34] and our preliminary calculations show that the blurring should be reduced greatly by introducing an appropriate deconvolution during data processing. The PSF deconvolution algorithm will be added to the new version of our STEMDIFF package. The deposition of nanocrystals onto a thinner carbon layer is expected to improve the results even further. The optimization of scanning parameters, such as beam intensity, spot size, and dwell time, could help as well. Systematic investigation of these effects is a subject of our ongoing research.

The second limitation is an intrinsic property of the 4D-STEM/PNBD method: by combining all diffractograms into one composite diffraction pattern, we lose the local information about the orientation of individual crystallites. Nevertheless, the method is still able to probe quite small areas on the investigated specimens. Considering that it is perfectly enough to collect around 8000 data files even for samples containing quite small crystallites (as evidenced for sample TbF_3_, Figure 4), the scanned area can be as small as 5 × 5 μm (or perhaps even smaller, depending on the specimen composition, morphology, thickness, etc.).

The third limitation is the maximum thickness of the samples that can be studied by the 4D-STEM/PNBD method. The situation is analogous to TEM/SAED: the crystals should be thin enough so that a sufficient number of diffracted electrons penetrate through the specimen. The penetration depth of electrons into a specimen decreases with the accelerating voltage. The maximum accelerating voltage in a great majority of SEM microscopes is 30 kV (this accelerating voltage was also used in our study), while the accelerating voltages in typical TEM microscopes range from 60 to 300 kV. Consequently, the thickness of the nanocrystals for 4D-STEM/PNBD should be even lower than in the case of TEM/SAED, but our results suggested that this limitation is not critical. The small- and medium-sized nanoparticles (TbF_3_ nanocrystals and Au nano-islands) yielded good results. Even the largest nanocrystals (NaYF_4_, size > 100 nm, thickness > 50 nm) exhibited diffraction intensity sufficient for further processing, although the diffraction peaks were close to the detection limit, as illustrated in Figure 9c.

### 4.3. Advantages and Future Perspective of the 4D-STEM/PNBD Method

The primary advantage of the 4D-STEM/PNBD method is its simplicity. The method converts a 4D-STEM data cube to one powder electron diffraction pattern. The conversion is quite straightforward and requires just a few Python scripts that were introduced in this work. The processing of the resulting powder diffraction patterns is much easier than the processing of the individual diffractograms. This makes the method accessible to common users of SEM microscopes that do not need deep crystallographic knowledge.

The second important advantage of the 4D-STEM/PNBD method consists in its low hardware and software requirements. It requires just one additional piece of hardware—a pixelated STEM detector—that can be mounted into any modern SEM microscope with a suitable free port. The size of a retractable pixelated STEM detector based on DED technology is quite comparable to a conventional STEM detector. There is no need to install bulky hardware such as CCD or CMOS cameras coupled with scintillators inside or outside the SEM microscope chamber. Software requirements are even lower: the basic software that controls the pixelated STEM detector is perfectly sufficient for data collection. The user just installs freeware Python programming language distribution that contains the standard scientific modules and downloads a set of simple Python scripts that are available upon request from the first author in the form of the STEMDIFF package. The usage of STEMDIFF scripts is very easy, as described in this contribution.

We have demonstrated that an arbitrary SEM microscope with a pixelated STEM detector can be converted to a more universal tool that enables easy and fast characterization of nanocrystal structures by means of the 4D-STEM/PNBD method. In this work, the method was introduced and verified for various nanocrystalline particles deposited on a thin carbon film. Nevertheless, in the near future, the method can be further improved and/or applied to other types of specimens. The key improvement of the method would be the increase in the intensity and resolution of diffraction peaks. The most promising way seems to be the introduction of a PSF deconvolution algorithm in the STEMDIFF package, as discussed in the previous section. Possible hardware improvements comprise positioning the detector further from the specimen holder or increasing the pixelated detector resolution. The 4D-STEM/PNBD method could also be applied to another, very interesting type of specimen–ultrathin sections prepared by means of focused ion beam (FIB). In fact, the microscope used in this study (Helios G4; Thermo Fisher Scientific) was a dual-beam microscope containing both electron and ion beam columns. The combination of FIB and pixelated STEM detector in one device enables to prepare thin lamellae from bulk specimens and characterize them routinely, by means of the fast and straightforward 4D-STEM/PNBD method.

## Figures and Tables

**Figure 1 nanomaterials-11-00962-f001:**
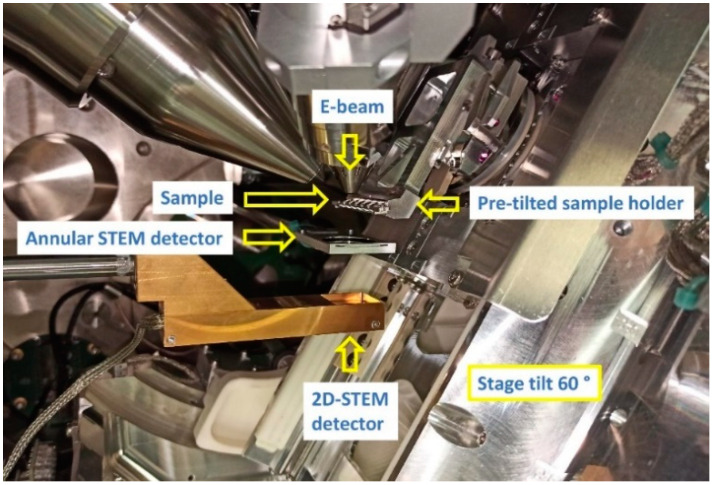
In-chamber situation of the FIB-SEM (Focused ion beam SEM) Helios during 4D-STEM/PNBD data collection: The annular STEM (scanning transmission electron microscopy) detector is used for precise beam alignment and imaging of the 4D dataset sample area. During 4D-STEM dataset acquisition, the annular STEM detector is retracted.

**Figure 2 nanomaterials-11-00962-f002:**
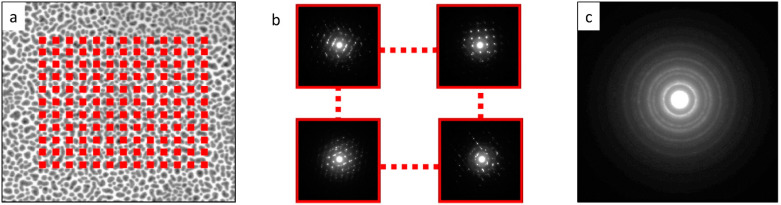
Principle of 4D-STEM/PNBD method: (**a**) Standard STEM/BF (bright-field) image of a sample with planed scanning matrix (represented schematically by red points). (**b**) Diffraction patterns captured from the individual locations (red points in (**a**)). (**c**) Powder diffraction pattern obtained by summation of all individual diffraction patterns, showing diffraction rings that are typical of polycrystalline materials. All images come from the first testing sample (Au nano-islands).

**Figure 3 nanomaterials-11-00962-f003:**
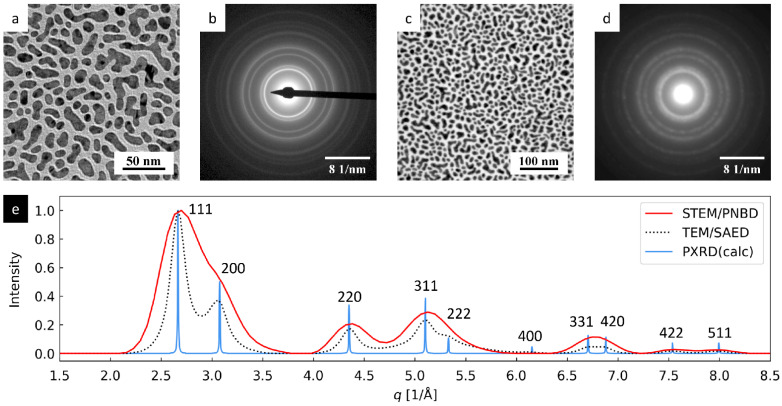
4D-STEM/PNBD results for Au nano-islands and their comparison with TEM and PXRD (powder X-ray diffraction) results: (**a**) TEM/BF image, (**b**) TEM/SAED diffraction pattern, (**c**) STEM/BF image, (**d**) 4D-STEM/PNBD diffraction pattern, and (**e**) comparison of radially averaged results from 4D-STEM/PNBD (red line), TEM/SAED (black dotted line), and theoretically calculated PXRD diffraction pattern of Au.

**Figure 4 nanomaterials-11-00962-f004:**
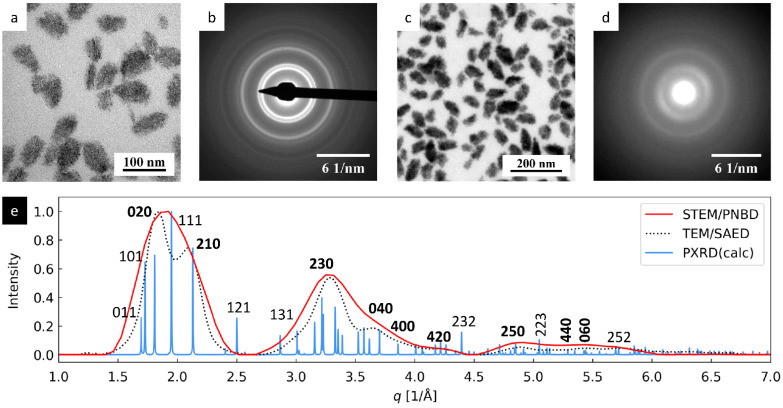
4D-STEM/PNBD results for TbF_3_ nanocrystals and their comparison with TEM and PXRD results: (**a**) TEM/BF image, (**b**) TEM/SAED diffraction pattern, (**c**) STEM/BF image, (**d**) 4D-STEM/PNBD diffraction pattern, and (**e**) comparison of radially averaged results from 4D-STEM/PNBD (red line), TEM/SAED (black dotted line), and theoretically calculated PXRD diffraction pattern of TbF_3_. The *hk0* diffraction peaks, which were stronger in TEM/SAED and 4D-STEM/PNBD than in PXRD due to preferred orientation of nanocrystals, are marked with bold font.

**Figure 5 nanomaterials-11-00962-f005:**
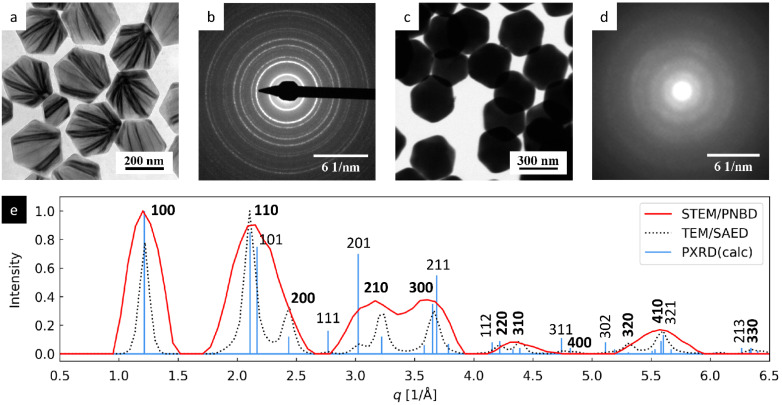
4D-STEM/PNBD results for NaYF_4_ nanocrystals and their comparison with TEM and PXRD results: (**a**) TEM/BF image, (**b**) TEM/SAED diffraction pattern, (**c**) STEM/BF image, (**d**) 4D-STEM/PNBD diffraction pattern, and (**e**) comparison of radially averaged results from 4D-STEM/PNBD (red line), TEM/SAED (black dotted line), and theoretically calculated PXRD diffraction pattern of NaYF_4_. The *hk0* diffraction peaks, which were stronger in TEM/SAED and 4D-STEM/PNBD than in PXRD due to preferred orientation of nanocrystals, are marked with bold font.

**Figure 6 nanomaterials-11-00962-f006:**
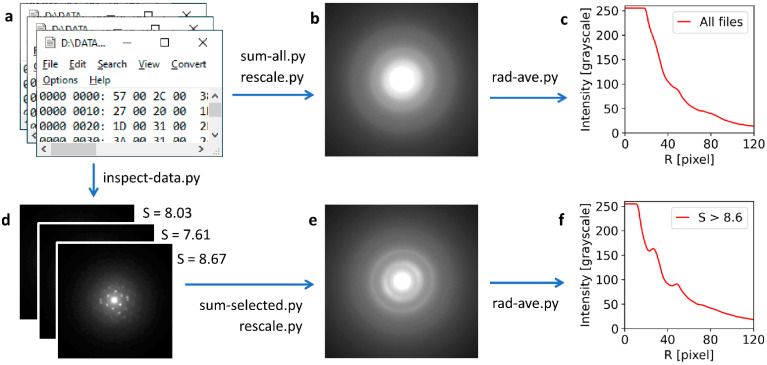
The processing of 4D-STEM/PNBD data by means of the STEMDIFF program package. The data come from the TbF_3_ dataset. (**a**) Input binary files containing raw data from pixelated detector in the form of 256 × 256 array, (**b**) the composite powder diffractogram created by the summation of all files by means of scripts *sum-all.py* and *rescale.py*, (**c**) the radially averaged intensity, created by script *rad-ave.py*, (**d**) the individual diffractograms with the corresponding calculated values of Shannon entropy, *S*, which can be gradually visualized by means of script *inspect-data.py*, (**e**) the composite powder diffractogram created by the summation of the diffractograms with high values of Shannon entropy (here: *S* > 8.6), created by scripts *sum-selected.py* and *rescale.py*, and (**f**) the radially averaged intensity, created by script *rad-ave.py*.

**Figure 7 nanomaterials-11-00962-f007:**
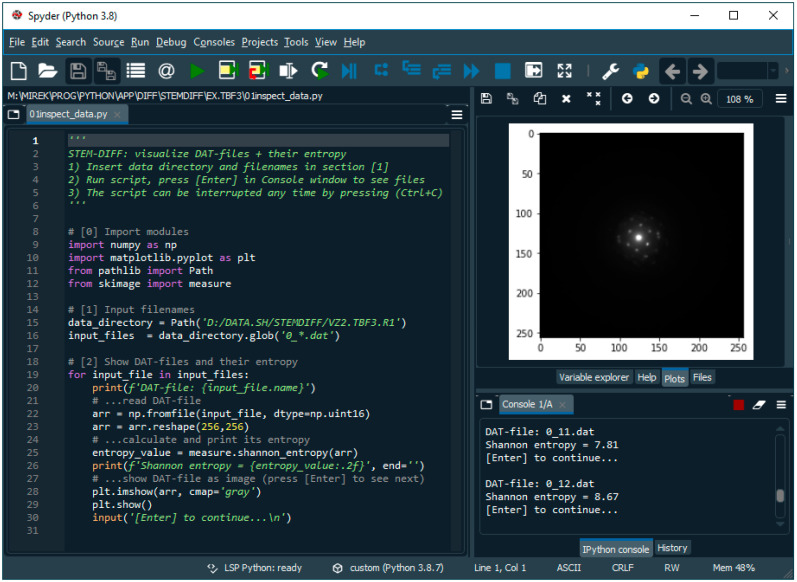
User interface of program package STEMDIFF is based on Spyder (freeware Scientific Python Development Environment). Selected script (here: *inspect-data.py*) is opened in the editor window (leftmost window) and slightly modified according to the description in its header (here we just define the names of input files). After running the script (Menu → Run), we can see image outputs (here: individual diffraction patterns) in the plot window (upper right) and numerical outputs (here: current file name and corresponding value of Shannon entropy) in the console window (lower right).

**Figure 8 nanomaterials-11-00962-f008:**
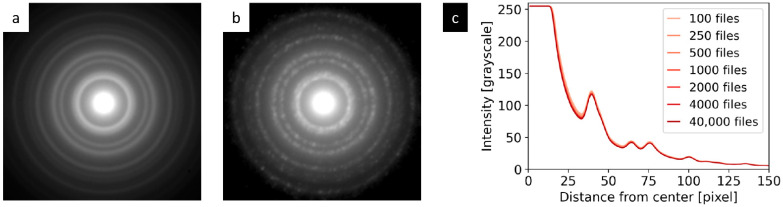
The influence of the dataset size on the quality of the final 4D-STEM/PNBD diffraction pattern of Au: (**a**) The initial 4D-STEM/PNBD diffraction pattern obtained by the summation of all 40,000 data files (i.e., full dataset *Au big* in Table 1), (**b**) the diffraction pattern obtained by the summation of just 100 files from the same dataset, and (**c**) the radial intensity profiles as a function of the dataset size. The contrast of the diffraction patterns (**a**,**b**) was increased for better visualization, while the radial profiles (**c**) show real intensities corresponding to unmodified images.

**Figure 9 nanomaterials-11-00962-f009:**
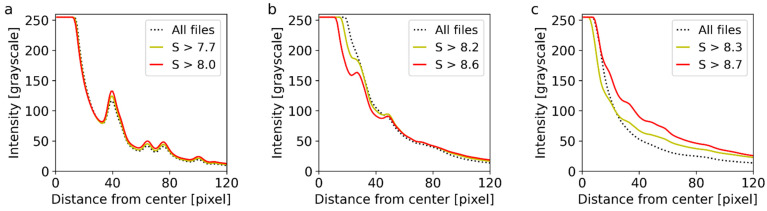
The influence of filtering on the final 4D-STEM/PNBD diffractogram of: (**a**) Au, (**b**) TbF_3_, and (**c**) NaYF_4_. In all graphs, the dotted line represents radially averaged intensity from the full dataset (Au big, TbF_3_, or NaYF_4_, Table 1), the yellow line represents the intensity after filtering out files containing just a few weak diffraction spots, the red line represents the intensity after keeping only the files with strong diffraction spots, and *S* stands for Shannon entropy value, as explained in Section 3.2.

**Table 1 nanomaterials-11-00962-t001:** Data acquisition settings.

Dataset ID	Step ^1^ (nm)	Scanning Matrix	HFW ^2^ [µm]	WD ^3^ [mm]	No. of Locations	Total no. of Files	Duration [h:m:s] ^4^
Au small	20	50 × 40	5	5.4	1	2000	0:05:30
Au big	20	200 × 200	5	5.4	1	40,000	1:52:00
TbF_3_	50	100 × 80	5	4.6	1	8000	0:22:00
NaYF_4_	200	50 × 40	10	3.0	10	20,000	0:55:00

^1^ Step = scanning step = distance between two scanning points^. 2^ HWF = Horizontal field width = real width of the STEM/BF image ^3^ WD = working distance, i.e. the distance between pole piece and the specimen ^4^ [h:m:s] = hours, minutes and seconds.^.^

## Data Availability

All data (STEM and TEM micrographs, 4D-STEM data cubes) and scripts used for 4D data processing (STEMDIFF package) are available upon request from the first author (M.S.).

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
