# Peer review of "Powder Nano-Beam Diffraction in Scanning Electron Microscope: Fast and Simple Method for Analysis of Nanoparticle Crystal Structure"

_nanomaterials, 2021, doi:10.3390/nano11040962_

Round 1

Reviewer 1 Report

The manuscript "Powder Nano-Beam Diffraction in Scanning Electron Microscope: Fast and Simple Method for Analysis of Nanoparticle Crystal Structure" shows a method to record electron diffraction patterns in transmission using a scanning electron microscope equipped with a pixelated detector below the sample. The analysis includes a simple way of filtering, and finally averaging of diffraction patterns in order to obtain data that be analyzed in the same way as conventional powder diffraction patterns.

The main benefit of the new method is its simplicity, and in particular it could be useful if no transmission electron microscope is available. Another advantage is some reduction of background noise by filtering the individual exposures prior to averaging. I wouldn't expect new insights with this method that could not be obtained by other means. But still, the paper can be useful to speed up standard analysis tasks, and hence it could be published.

Overall, the paper is well written and understandable. I have a few technical comments that could be adressed in a minor revision:
1) Instead of scanning a highly focused beam and averaging a large number of diffraction patterns, couldn't one just spread the beam over the desired area of the sample (e.g. by defocusing the SEM) and record a standard powder diffraction pattern?  If this is possible, it might be useful to show a comparison.
2) Some further description of the detector should be given (even if it is given in a reference, a summary is useful). For example, its dynamic range, quantum efficiency with 30keV electrons, etc.
3) The authors frequently use the term "diffractions", which seems unusual to me. I suggest to replace this word by "diffraction peaks" or other suited terms.
4) Typos that I noticed (probably not a complete list): Fig. 8 caption "inal" (initial?), and Fig7 caption "patters" (patterns).
5) I suggest to provide the scripts in a supplementary information, instead of on request from the author.

Author Response

Overall, the paper is well written and understandable. I have a few technical comments that could be addressed in a minor revision:

1) Instead of scanning a highly focused beam and averaging a large number of diffraction patterns, couldn't one just spread the beam over the desired area of the sample (e.g. by defocusing the SEM) and record a standard powder diffraction pattern?  If this is possible, it might be useful to show a comparison.

ANSWER: In SEM the beam is always focused to a small spot, whose size can be changed in a limited range. Even if the beam was defocused to a very broad spot in some non-standard way, the individual paths of the electrons would be divergent and, as a result, the standard diffraction pattern could not be obtained. Nevertheless, the reviewer is right that the spot size could influence the quality of the results somehow. As mentioned in section 4.2. (at the end of the second paragraph), in our future work we are going to investigate the effect of scanning parameters (including the spot size) on the quality of the results.

2) Some further description of the detector should be given (even if it is given in a reference, a summary is useful). For example, its dynamic range, quantum efficiency with 30keV electrons, etc.

ANSWER: The description of the detector was improved. We inserted the information about type of detector (Si sensor) and declared quantum efficiency at 30 keV (> 90%) to the Experimental section (section 2.4.1, 2nd paragraph).

3) The authors frequently use the term "diffractions", which seems unusual to me. I suggest to replace this word by "diffraction peaks" or other suited terms.

ANSWER: The reviewer is right that the usage of the term [diffraction] instead of more verbose and more precise terms such as [diffraction peak] or [diffraction spot] or [diffraction ring] is not standard. We corrected this throughout the manuscript (the changes were made with MS Word Revision tools so that they could be easily recognized).

4) Typos that I noticed (probably not a complete list): Fig. 8 caption "inal" (initial?), and Fig7 caption "patters" (patterns).

ANSWER: Both typos were corrected. We thank the reviewer for careful reading. We also checked the rest of the text with [MS Word – Revisions – Language tools] and we believe that the revised manuscript is now correct from the point of view of typographic errors.

5) I suggest to provide the scripts in a supplementary information, instead of on request from the author.

ANSWER: We decided to offer the scripts at request to the first author due to the fact that the method is quite new and the scripts are under continuous development. In fact, the current scripts are newer, slightly improved, and containing better documentation than the original scripts used for the calculations in the submitted manuscript. Therefore, if someone asks for the scripts, he or she will receive the very recent version.  In our next contribution (under preparation) we are going to put the scripts on www and update them continuously.

Reviewer 2 Report

In the manuscript under review, Miroslav Slouf and co-authors elaborate on a possible development of a SerialED method for SEM instead of TEM, assuming the acquisition of individual diffraction snapshots from a rectangular grid covering some densely packed nanoparticles, each in a single orientation, followed by the data processing using a STEMDIFF package also developed by the authors. The data processing includes but not limited to a summing of the intensities in all frames and a calculation of the radially averaged intensity, and the resulting ring pattern contains d-spacings which are typical for a certain crystal structure. This manuscript is submitted as a contribution to the “Microscopy of Nanomaterials” special issue, an in my opinion it fits very well to the announced scope, and can be published after a minor revision. However, I recommend the authors to address a following minor comments and concerns:

§3.3.2 Dataset filtering: For the TbF3 and NaYF4 nanoparticles the entropy based filtering improves FWHM of diffraction peaks obtained from the rings, as shown in Fig. 9. At the same time, the high hkl reflections which are typically weaker than the low-hkl ones, also quite likely are the subject to be removed by the filtering, hindering the ring pattern resolution = min achievable d-spacing which is distinguishable from the background. Could the authors suggest the optimal protocol or a criteria list for the filtering, since the data in many cases would obviously be biased by this procedure?

§4.2 Current limitations: SEM mainly remains a method of a bulk analysis. Please address the thickness issue – how thin the specimen should be in order to get the crystallographic info about it in a standard 30 keV SEM by your method?

Line 61: Scanning beam is thin = the probe size is small or maybe the specimen is thin? please reformulate this sentence. One related question – what is the minimal probe size achievable in a SEM at your conditions?

Lines 174-178: I believe that the sum of individual diff patterns in your case is rather a ring diffraction pattern than a powder diffraction pattern, based on a very small size of the scattering domains.

Author Response

This manuscript is submitted as a contribution to the “Microscopy of Nanomaterials” special issue, an in my opinion it fits very well to the announced scope, and can be published after a minor revision. However, I recommend the authors to address a following minor comments and concerns:

  • 3.3.2 Dataset filtering: For the TbF3 and NaYF4 nanoparticles the entropy-based filtering improves FWHM of diffraction peaks obtained from the rings, as shown in Fig. 9. At the same time, the high hkl reflections which are typically weaker than the low-hkl ones, also quite likely are the subject to be removed by the filtering, hindering the ring pattern resolution = min achievable d-spacing which is distinguishable from the background. Could the authors suggest the optimal protocol or a criteria list for the filtering, since the data in many cases would obviously be biased by this procedure?

ANSWER: The results showed that the entropy-based filtering in fact increased the intensity of all peaks, regardless of their hkl indexes. This is documented in Figure 9: the diffraction peaks were enhanced in the whole range for all three studied systems (strongly diffracting Au nanoislands, small TbF3 nanocrystals, and large NaYF4 nanocrystals). At the moment we have no optimal protocol for filtering – this requires more experiments, but the results suggest that the results are definitely not biased by the filtering procedure. Moreover, if the results were biased, the user could notice this by comparing original (unfiltered) dataset (black dotted curves in Fig. 9) with filtered datasets (yellow and red lines in Fig. 9). Last but not the least, the comparison of our 4D-STEM/PNBD data with TEM/SAED experiment and PXRD calculation (Figs. 3-5) documented that the filtered datasets are in very good agreement with both independent experimental results (TEM/SAED) and theoretical prediction (PXRD). We added one more paragraph at the end of section 3.3.2 of the revised manuscripts, which briefly summarizes the above-discussed facts.

  • 4.2 Current limitations: SEM mainly remains a method of a bulk analysis. Please address the thickness issue – how thin the specimen should be in order to get the crystallographic info about it in a standard 30 keV SEM by your method?

ANSWER: Our results suggest that the maximum thickness of the specimens suitable for 4D-STEM/PNBD is slightly above 50 nm. This estimate is based on the last sample, NaYF4 crystals, whose diameter was >100 nm and thickness >50 nm. The diffraction intensities from these relatively thick nanocrystals were at the edge of detection limit of the method, as evidenced in Figure 9c. We added one more paragraph to section 4.2 (the last paragraph of the section), in which we describe this additional limitation of 4D-STEM/PNBD method.

Moreover, our preliminary calculation showed that the estimated mean free path, λ, of 30 keV electrons in NaYF4 was 14 nm. The experimentally determined thickness of the crystals was above 50 nm and for the thickest crystals even up to 100 nm. As the diffraction intensities were very low, perhaps at the edge of detection using our technique, this example suggested that the mean free path of electrons at given crystals should be 7x lower than the thickness of the crystals (100/14 = 7.7). More detailed evaluation of this phenomenon is s subject of our ongoing research.

  • Line 61: Scanning beam is thin = the probe size is small or maybe the specimen is thin? please reformulate this sentence. One related question – what is the minimal probe size achievable in a SEM at your conditions?

ANSWER: We wanted to say that the probe size is small. We modified the text so that this was clear. The changed sentence now reads: “If the scanning beam is thin narrow (and not convergent), the probe size is small and the recorded patterns correspond to nanobeam diffraction (NBD) known from TEM (we note that the classical TEM/NBD method is performed with at fixed beam position [10]).”

Answer to the related question: Minimal probe size in our microscope at given conditions (i.e. in high-resolution mode) is slightly below 1 nm.

  • Lines 174-178: I believe that the sum of individual diff patterns in your case is rather a ring diffraction pattern than a powder diffraction pattern, based on a very small size of the scattering domains.

ANSWER: The terms [powder diffraction pattern] and [ring diffraction pattern] are equivalent in the sense that powders (i.e. polycrystalline materials) yields ring diffraction patterns. That is why the terms powder diffraction pattern and polycrystalline diffraction pattern and ring diffraction pattern are used interchangeably. See, for example [https://www.intechopen.com/books/modern-electron-microscopy-in-physical-and-life-sciences/electron-diffraction]. The figure caption (lines 174-178 in the original manuscript) was slightly modified so that this was clearer.